# Nutritional Status and Poverty Condition Are Associated with Depression in Preschoolers

**DOI:** 10.3390/children10050835

**Published:** 2023-05-04

**Authors:** Betsabé Jiménez-Ceballos, Erick Martínez-Herrera, María Esther Ocharan-Hernández, Christian Guerra-Araiza, Eunice D. Farfán García, Uriel Emiliano Muñoz-Ramírez, Claudia Erika Fuentes-Venado, Rodolfo Pinto-Almazán

**Affiliations:** 1Clínica de Trastornos de Sueño, Universidad Autónoma Metropolitana Unidad Iztapalapa UAM-I, Av. San Rafael Atlixco 186, Leyes de Reforma 1ra Secc, Iztapalapa, Ciudad de Mexico 09340, Mexico; 2Sección de Estudios de Posgrado e Investigación, Escuela Superior de Medicina, Instituto Politécnico Nacional, Plan de San Luis y Díaz Mirón, Ciudad de Mexico 11340, Mexico; 3Efficiency, Quality, and Costs in Health Services Research Group (EFISALUD), Galicia Sur Health Research Institute (IIS Galicia Sur), SERGAS-UVIGO, 36213 Vigo, Spain; 4Unidad de Investigación Médica en Farmacología, Centro Médico Nacional Siglo XXI, IMSS, Ciudad de Mexico 06720, Mexico; 5Non-Communicable Disease Research Group, Facultad Mexicana de Medicina, Universidad la Salle-México, las Fuentes 17, Tlalpan Centro I, Tlalpan, Mexico City 14000, Mexico; 6Servicio de Medicina Física y Rehabilitación, Hospital General de Zona No 197 IMSS, Texcoco 56108, Mexico

**Keywords:** depression, preschool, nutritional status, malnutrition, sociodemographic factors, access to healthcare

## Abstract

Consensus has been reached that symptoms of depression can begin as early as preschool. Nevertheless, only few studies have associated environmental (malnutrition) and social factors (poverty condition, access to health systems, etc.) to the onset of depression in preschoolers. The aim of this study was to explore possible associations between malnutrition (underweight, overweight/obesity), poverty status (home quality, overcrowding), access to healthcare systems and the presence of depressive symptoms in the preschoolers of a semi-rural community. In total, 695 children between 3 and 6 years from the municipality of Chiconcuac, Mexico were evaluated for symptoms of depression with the Preschool Depression Scale for Teachers (ESDM 3-6). Additionally, they were assessed for nutritional status and divided into three groups (low weight, normal weight, overweight/obesity), and their parents were asked to fill out a social demographic questionnaire. Malnutrition status OR = 2.702, 95% CI [1.771–4.145]; UW OR = 4.768, 95% CI [2.570–8.795] and OW/OB OR = 1.959, 95% CI [1.175–3.324]; poverty condition per se OR = 1.779, 95% CI [0.9911–2.630]; housing quality OR = 2.020, 95% CI [0.9606–2.659] and overcrowding = 1.619, 95% CI [0.8989–4.433] were associated to a greater risk for children to show depressive symptoms (DS). Access to healthcare was negatively related with the risk of presenting DS (OR = 0.660, 95% CI [0.3130 to 1.360]). Social and environmental factors such as malnutrition, home quality and overcrowding may increase the risk of presenting DS as soon as in preschool.

## 1. Introduction

Depression is a pathology characterized by sadness and irritability, fatigue, negative self-perception, changes in appetite, loss of concentration, and in the worst cases, anhedonia and recurring thoughts about suicide and death [1,2]. 

Recently, researchers have centered their studies in characterizing and validating proper depression diagnostic measurements for preschoolers [3]. According to the DSM-5, there are no differences in the clinical presentation of depression between adults and children, but the diagnosis of early depression must also consider intense irritability associated to anhedonia, extreme guilt, shame and/or social isolation [4]. Depressed preschoolers may show behavioral changes in play, social interests, or sleep disturbances. In fact, even in these early ages, some children have death and suicide in mind and might hurt themselves as previously reported by Luby et al. [5]. The Preschool Depression Scale for Teachers (ESDM 3-6) is a screening validated instrument for epidemiological studies designed to identify depressive symptoms (DS) in preschoolers from information provided by teachers. ESDM 3-6 questions cover detectable mood disorders in early depression to report it as soon as possible and avoid chronification [6,7]. In Latin America and Mexico, this instrument has been validated identifying between 0 and 30% of DS within the Latino population [8,9,10]. Notably, this tool provides an objective vision of the pathology from a different point of view, outside the family environment. Thus, this might be helpful for the diagnosis as it is common that family members are biased when asked about something that ruins that optimal environment, so having additional referral becomes relevant in studies such as this [8,9,10].

Numerous authors agree that the environment in which the child develops is important to be analyzed in order to obtain a whole comprehensive understanding of the child’s health [11]. Scientific evidence has stated that both family and individual factors can vulnerate the mental health of children leading to symptoms and disorders of high-functioning depression. The behavior of the child may be influenced by family predisposition, psychosocial (stress and depression of parents, parents’ education, family type), biological/physiological (nutritional, immunological, genetics, pollution, or even smoking habits of the parents) and socioeconomical factors (family income, overcrowding and housing) [12,13].

Regarding biological sources, reports indicate that sex differences reveal higher prevalence of depression in women than men (2:1) from adolescence to adulthood [14]. Unlike adolescents and adults, no sex-related differences have been associated to depression in preschoolers [15,16]. Another feature to be taken into account is that depression at this age is highly associated with other co-morbidities [1,2]. Nutritional status (malnutrition extremes) has been associated with depression in adolescents and adults as well [12,16]. 

In terms of social basis, financial difficulties such as poverty, including poor housing quality and overcrowding, have been linked to an increased risk of DS in infancy and might produce immediate and long-term outcomes [17,18,19]. Moreover, economically underprivileged children have increased risks for suffering externalizing and internalizing behavioral problems [20]. It is worth mentioning that poverty refers not only to a low family income, but also to deprivation of healthcare and social services, absence of education, and the basic services for household and/or food access, among other factors [21,22,23].

Chiconcuac de Juarez is a municipality in the State of Mexico that is reported to have 50% of its population living in moderate poverty, 20% suffer vulnerability due to social deprivation, and there are no data about the lack of quality and housing spaces [24,25]. Since the socio-emotional domain in child development is mainly affected by nutrition and poverty, the high malnutrition and poverty status of the children living at Chiconcuac de Juarez might be increasing the DS of kids in the municipality when compared to normal-weight children with no poverty condition. 

Considering all the above, this study aimed to examine associations between malnutrition, poverty status and access to healthcare systems with regard to the presence of symptoms of depression in the preschool population of a semi-rural community of the State of Mexico.

## 2. Materials and Methods

### 2.1. Participants Selection

The research was approved by the Research and Ethics Committees of the Hospital Regional de Alta Especialidad de Ixtapaluca (HRAEI) (NR-027-2017) and was carried out following the Helsinki Declaration. The inclusion criteria for the study were: healthy preschoolers (3 to 6 years old) studying in public institutes at Chiconcuac de Juarez, a semirural municipality of the State of Mexico, and whose parents or tutors had signed the informed consent. A total of 981 children from 8 kindergartens were asked to participate in the study. Parents or guardians were invited to attend school meetings to inform them about the purpose of the study and to sign an informed consent form in case of acceptance. After the meeting, almost 71% (695 children) of the kindergarten students in the municipality were included in the study (380 boys and 315 girls). The age range of the studied population was 3 to 6 years, with an average of 4.63 ± 0.83. Subsequently, children’s medical history and sociodemographic questionnaire were distributed to their parents to be filled out as well. 

#### Poverty Condition Determination 

The sociodemographic questionnaire consisted of inquiries about the characteristics of the family, number of persons living at the same home, the education level of the parents, the conditions of the family’s housing (whether basic home services were available, the number of rooms in the house, as well as the characteristics of floors, ceilings, and building materials) and the accessibility to health services. 

With the information obtained from the sociodemographic questionnaire, we validated the determination of poverty condition (housing quality and overcrowding) with National Council for the Evaluation of Social Development Policy (Coneval) criteria, verifying whether lack of housing quality, overcrowding, or both were present [24].

As for the quality of the house, parameters included materials of the house such as dirt floors, roof materials such as cardboard sheet or waste, as well as the material of the walls of the house such as mud or bahareque, cane, bamboo or palm, cardboard, metal or asbestos sheets, or even waste material.

Additionally, the overcrowding of each household was calculated; the ratio of persons per room (overcrowding) is greater than 2.5.

### 2.2. Anthropometric Determination

After the parents signed the informed consent form to be considered in the study, anthropometric data were taken from all the participants in order to determine the nutritional group to which they belonged. Trained and standardized staff performed anthropometric measurements of the preschoolers who were accompanied by their parents or guardians at all times. Using a scale with stadiometer (Seca 700; Hamburg, Deutschland), height (cm) and weight (kg) values were obtained from the participants. The values were converted into Z-scores and consequently input in the AnthroPlus software (version 3.2.2, 2011, Geneva, Switzerland) based on WHO guidelines. Children were divided into three groups according to the Z-scores: underweight (UW) (Z-score < −2), normal weight (NW) (Z-score −1 to 1), overweight plus obesity (OW/OB) (Z-score > 2) [26]. 

The majority of the children were grouped into normal weight group (n = 529, 76.12%) followed by overweight (OW)/obesity (OB) (n = 113, 16.26%) and underweight groups (UW) (n = 53 children, 7.63%).

### 2.3. ESDM 3-6

As the anthropometric determination was conducted, educators were requested to complete the ESDM 3-6 scale for each of their students. Fifty teachers filled out the 695 ESDM 3-6 formats in a range of 10 to 15 per instructor (mean = 13.9 ± 5.5). The ESDM 3-6 consists of 19 items with three possible responses per question: “almost never”, “sometimes” or “almost always”. As mentioned, ESDM 3-6 scale covers the mood disorders most detectable by educators. The score for each item is between 1 and 3. The purpose of the questionnaire is to identify possible dysphoric mood, the socialization capacity with educator and peers, the attitude of the child towards school, irritability caused by certain situations, school performance (concentration, attention and care in their activities), and also the level of physical activity (interest and participation).

The teachers answered the questions depending on the regularity with which, in their opinion, the behavior is presented and scored on the scale. The scales were only considered valid when all their items were answered unequivocally [6,7].

### 2.4. Statistical Analysis

Continuous data were expressed as mean ± standard deviation and categorical data in proportion. In categorical variables, comparison between kids with and without DS was performed with χ^2^. Binary logistic regression analyses were utilized to evaluate the relationship between malnutrition and depression (IC95%, *p* ≤ 0.05). Possible confounding variables in these interactions were integrated in the model, including socio-demographic factors and nutritional status. GraphPad Prism 8.0 (GraphPad Software, LLC, Boston, MA, USA) and IBM SPSS Statistics v28.0 (IBM Corporation, New York, NY, USA) were utilized for the statistical analysis. 

## 3. Results

### 3.1. Demographic and Anthropometric Data

Table 1 shows the demographic and nutritional variables of the population studied. Out of our sample, 109 (68 boys and 41 girls) were reported to show DS, corresponding to 15.68% of the overall population (17.89% of boys and 13.02% of girls). Regarding the male group with DS, it was observed that 19.12% (n = 13) were from UW group, 23.53% (n = 16) were from the OW/OB group and 57.35% were from the normal-weight group. The female group with DS (n = 41) consisted of 19.51% (n = 8) from the UW group, 19.51% (n = 8) from the OW/OB group and 57.35% from the normal-weight group (Table 1). 

Chi-Square Test of Independence was performed to determine whether sex was associated with depression, nutritional status (underweight UW alone, overweight OW/obesity OB alone and UW+OW/OB). No association was observed between sex and any of the malnutritional status groups, neither with poverty condition nor access to healthcare. Although there was a trend of association between sex and DS, no significant relationship was observed (Table 2).

### 3.2. Depression, Nutritional Status, Poverty and Access to Healthcare

Odds ratios for the relationships between nutritional status, poverty condition (housing quality/overcrowding), family type, parents’ education and access to healthcare and DS are summarized with the univariate analysis in Table 3 and the multivariate analysis in Table 4.

On unadjusted analysis, in the malnutrition status in general, 12.10% of the normal-weight children had DS compared with the 27.11% of children who were malnourished and with DS (OR = 2.702, 95% CI [1.771–4.145]; n = 45). Subgroup analysis reported 39.62% of the UW children (OR = 4.768, 95% CI [2.570–8.795]; *p* < 0.001) and 21.39% of the OW/OB with DS (OR = 1.959, 95% CI [1.175–3.324] *p* < 0.001). 

In terms of poverty condition per se, higher risk of DS was observed in children living in poverty condition (22.46%) compared to 14.00% of preschoolers with absence of poverty and with DS (OR = 1.779, 95% CI [0.9911–2.630]; *p* < 0.05). Regarding housing quality, 36.00% of the children with absence of quality housing presented DS compared to the 15.13% of the children with good housing quality and DS (OR = 2.020, 95% CI [0.9606–2.659]; *p* < 0.05). Regarding overcrowding, the risk of presenting DS increased in children living in overcrowded places (21.62%) compared to the children that did not (OR = 1.619, 95% CI [0.8989–4.433]; *p* < 0.05). 

Regarding family type, although biparental families seem to represent a lower risk for children to show DS (13.91%) when compared to children from monoparental families (27.91%), this was not statistically significant in the univariate analysis (OR = 0.417, 95% CI [0.247–0.705]; *p* < 0.19). 

Regarding the parents´ level of education, 7.53% of the kids whose mothers had more than 9 years of study had DS compared to 24.92% of children with mothers reporting less than 9 years of study (OR = 0.245; 95% CI [0.154–0.389]; *p* < 0.001). As for their fathers, 6.82% of children with fathers who had more than 9 years of education showed DS compared to 22.72% of children with fathers reporting less than 9 years of education (OR = 0.249; 95% CI [0.150–0.413]; *p* < 0.001). 

Interestingly, access to healthcare was negatively associated with the risk of presenting DS. This meant 10.7% of the kids who showed DS had the opportunity to receive health services, whereas 7.3% of children with DS did not (OR = 0.660, 95% CI [0.3130 to 1.360]; *p* < 0.05).

In multivariable regression analysis, after adjusting for sex, family type and parental education, the size of the effect of malnutrition on DS outcomes increased (OR = 7.490, 95 CI [1.224–3.402], *p* = 0.01) for UW group (OR = 9.480, 95 CI [1.527–6.719], *p* = 0.01) and for OW/OB (OR = 9.480, 95 CI [1.527–6.719], *p* = 0.01). In addition, the effect of poverty (OR = 4.378, 95 CI [1.039–3.197], *p* = 0.05) and of overcrowding (OR = 4.869, 95 CI [1.080–3.660], *p* = 0.05) increased while the effect of housing quality turned out non-significant (*p* = 0.77) 

Concerning healthcare access, the negative association determined before was no longer significant (*p* < 0.08), too. 

## 4. Discussion

The objective of this study was to determine the prevalence of children with DS and the association with their nutritional status, the poverty condition of families and access to health services. Unfortunately, we have demonstrated that the children living at Chiconcuac de Juarez with malnutrition status and in poverty have increased DS when compared to normal-weight children with no poverty condition. 

Key domains in childhood development are physical, cognitive, language and social–emotional development areas [25]. Childhood depression can be triggered by traumatic experiences such as losing loved ones or pets, moving to a new school, abuse/bullying, etc. [1,5]. In extreme cases, depression in preschoolers is related to thoughts or suicidal ideation/suicidal behaviors (SI/SB), and/or the possibility to hurt themselves, also called non-suicidal self-injurious behaviors (NSSI) [1,5]. 

The behavior of the child may be influenced by family predisposition, psychosocial (stress and depression of parents, parents’ education, family type), biological/physiological factors (nutritional, immunological, genetics, pollution, or even smoking habits of the parents) and socioeconomical basis (family income, overcrowding and housing) [12,13]. Galván et al. and Tevormina et al. emphasized that the environment where the child develops is important for the analysis of physiological factors (nutritional, immunological and genetic status) that ultimately influence the overall behavior of the child [12,13].

Latin America is a region with several significant health problems in childhood such as infections. Extreme malnutrition related to high poverty settings negatively impacts early childhood development. Until 2020, 8.8 million children were estimated to have malnutrition (4.8 million with undernutrition and 4 million with overweight or obesity) and almost 50% of children between 0 and 14 years lived in poverty conditions in this part of the world [21]. As for Mexico, in 2020, it was estimated that almost 20% of the children between 0 and 4 years old had UW, 8.4% had OW/OB and another 22% of had an increased risk of overweight [21,22]. Chiconcuac de Juarez is a semirural municipality in the State of Mexico with a huge proportion of population in moderate poverty, vulnerable to social deprivation. Nevertheless, the Coneval (National Council for the Evaluation of Social Development Policy) website lacks information on quality and housing spaces in the community [24]. On the other hand, WHO has highlighted the impact of depression among children and adolescents for the past decade. Since 2014, it has been identified as the main cause of illness and disability in boys and girls between 10 and 19 years of age. In 2019, 280 million people were living with depression, including 23 million children and adolescents [27]. More recently, in 2021, UNICEF´s central topic in the State of World´s Children report about promoting, protecting and caring for children´s mental health indicated that after the COVID-19 pandemic, around 1 to 5 young people between the ages of 15 and 24 feel depressed or have little interest in making plans. Among those, most at risk were the millions of children forced to leave their homes, scarred by conflict and other serious hardships, and deprived of access to education, protection, and help [28].

From a sociodemographic perspective, the etiological multifactoriality of depression reflects an interesting contrast between world regions. For example, in America, about 5% of the population was reported to have depression in 2015, and of this, 0.4% to 2.5% represented the children population, placing depression as the second most frequent pathology according to paidopsychiatrists [29].

In Latin American countries such as Chile, the total prevalence of depression between the ages of 4 and 18 reached 6.1%. Colombia and Mexico share very similar data, with an approximate prevalence of 2%. However, in the latter, affective disorders due to depression came to occupy the first place of care (25.7%) in a nationwide children’s psychiatric hospital [29].

On the other hand, data from the European Health Interview Survey collected between 2013 and 2015 indicated that 6.4% of the European population suffered from depression. These data were higher than those estimated by the WHO, which had calculated the prevalence of this pathology in the European region at 4.2%. By country, those with the highest prevalence were Iceland (with 10.3% of the population), Luxembourg (9.7%), Germany (9.2%) and Portugal (9.2%). The lowest prevalence rate of depression was reported in the Czech Republic (2.6%), Slovakia (2.6%), Lithuania (3%) and Croatia (3.2%). A very notable fact from the study, as can be seen, is that the countries with higher economic development and, therefore, in theory, with more healthcare resources, showed rates of depression up to four times higher than average [30]. Regarding the pediatric population, Spain reports highly representative statistics for the prevalence of major depressive disorder (MDD): 1.8% in 9-year-old children, 2.3% in 13- and 14-year-old adolescents, and 3.4% in 18-year-olds [31].

Despite the latest progress in understanding depression in school-aged children and adolescents, there is limited information about the prevalence of depression in children younger than 6 years of age. Domènech-Llaberia et al. designed and validated the ESDM 3-6 screening scale in a stratified randomized study in preschoolers and demonstrated its usefulness for screening preschool depressive symptomatology in epidemiological studies [6,7].

Domènech-Llaberia et al. reported a high frequency of preschoolers with DS (almost 1 in 7 children) assessed through the ESDM 3-6. Coincidentally, Domènech-Llaberia et al. reported the same percentage of children with DS (15.6%) as our research. In addition, out of their sample, 1.12% were diagnosed with MDD [7]. In our local context, in 2005, unlike what was observed in this study, Reyna Y. did not discover children with DS in her study conducted in 80 public preschools in the City of Zacatecas, State of Zacatecas, Mexico [9]. In turn, Perez F. conducted a study on a psychopedagogical proposal to reduce DS and promote language learning in the preschool population of a public kindergarten in the City of Cuernavaca, State of Morelos. During the initial evaluation, Perez F. reported that 30% of the preschool population studied suffered from DS associated with decreased language/communication skills, and that these data could be reversed after the narrative therapy proposed [10].

Concerning biological factors, we observed a high prevalence of children with malnutrition status (41.3%) who showed DS, from which it is worth mentioning that underweight children (both boys and girls) are most likely to develop DS. Childhood malnutrition induced by constant interruption or even by sporadic/discreet undernutrition of nutrients can produce developmental (physical size, fine/gross motor skills, cognitive growth, etc.) and behavioral (self-regulation, anxiety and depression) outcomes and deficits [32]. Prior reports on moderate–severe undernourished samples in early life ages demonstrate persistent cognitive and behavioral deficits when compared to those of peers without malnutrition histories [33]. More relevant is the fact that behaviors such as a four-fold decrease in attention symptoms in elementary school, certainly linked to early childhood undernourishment, predicted poor performance in high school entrance examination and DS at 11–17 years [34]. Various studies have consistently established that overweight and obese children and adolescents have a significantly higher risk of developing DS and even MDD compared with healthy controls [35]. As for the other side of malnutrition, Cañoles et al. reported in a population of Chilean preschoolers that obesity had no incidence/relationship with DS in preschoolers [8]. These results are consistent with ours from the multivariate analysis, where we could observe that obesity does not have a statistically significant impact when analyzing DS with other important cofactors. 

When associating nutritional status and gender, no statistical difference was observed regarding sex differences in preschool depression. Epidemiological reports in adults have reported a twofold increase in prevalence of females compared to males with depression, which is forecasted to increase sharply in adolescence [36,37]. Accordingly, Lewis et al. and Breslau et al. reported that females between 4 and 14 years were predisposed to an increasing prevalence of DS than males at the same age, and this predisposition becomes significant at the age of 12 years (5.2% in female versus 2.0% in male subjects, *p* < 0.0001) [14,38]. Interestingly, sex hormone variations are important during this period and throughout the menstrual cycle, gestation, labor, and menopause that take part crucially for some depressive syndromes [39]. With that in mind, it is understandable that preschoolers record no differences in DS prevalence between girls and boys [15,16]. According to the previous literature, boys and girls are equally affected by DS before adolescence, or boys are hardly favored [40,41,42]. The incidence of depression in both sexes has been estimated at 3% of the children that fulfill diagnostic criteria of MDD [43]. Nevertheless, it has been precisely reported that obese female children have a significantly higher odds of depression compared with normal-weight female children [44]. Because of misdetection, depression in preschool-aged children may be underestimated [45]. Domènech-Llaberia et al. reported that the DS analyzed by the ESDM 3-6 appeared in relation of 1:1 girls vs. boys [7]. However, other biological correlates proposed to validate the early-onset disorder have evidenced alterations in the hypothalamic–pituitary–adrenal axis, similar to those seen in adult depression, and increased rates of affective disorders in family members of depressed preschoolers as well [46,47]. With growing evidence of childhood depression in the first years of life (as early as preschool), some studies have recently suggested that gender difference in depression originates during childhood and grows in magnitude during adolescence. In youth, high levels of impairment, suicide attempts, conduct problems and poor academic functioning argue against a “wait and see” approach to clinical treatment of recent first-onset depression [38].

In addition, numerous factors such as preterm births or small products for gestational age, less responsive or unsensitive parents to child´s needs, less stimulating home environments and extreme poverty and food insecurity have been associated to nutritional insufficiencies in early childhood [32]. It has been also described that poverty is one of the main barriers that prevent Mexican children and adolescents from growing/developing their potential, since more than half of children and adolescents live in conditions of poverty [22].

In terms of poverty, the Mexican government had previously defined poverty in a multidimensional analysis considering aspects of the living environments of the population from three areas: economic well-being (needs associated with the goods and services that can be acquired through income), social rights (deficiencies in the use of their rights for community development) and territorial context (degree of social organization as well as other factors reflected in social development). Thus, a person can experience multidimensional poverty when they have insufficient income to acquire goods/services for their requirements, which can lead to a lack of education, access to healthcare services, social security, housing quality and spaces, and the basic services for household and/or food access [23].

Within the set of experiences of poverty, poor diet in both quality and quantity has been associated with mood disorders such as depression in children, particularly between ages 3.5 and 7 [15,16].

Likewise, it was observed that the poverty condition of families significantly increased (OR = 1.779; CI 95% 0.9911–2.630) the risk of suffering DS in the preschool population, and so did specific components such as housing quality (OR = 2.020; 95% CI 0.9606–2.659) and overcrowding (OR = 1.619; CI 95% 0.8989–4.433). Zhang et al. described that the lack of sufficient household income influences depression and inadequate stimulation of children contributes negatively to poor performance and preschool and school achievement. Additionally, the authors determined that poverty and all those risks by themselves entailed exacerbation of the so-called psychosocial risks, seriously harming the development of the child [48].

In concordance with our study, some experts have pointed out that a low socioeconomic status (SES) may confer risk for depression through higher levels of perceived and objective stress and cumulative environmental risk such as poor housing quality, noise pollution, and exposure to violence. Lower socioeconomic status has been associated with a host of negative outcomes including poorer general health and increased risk of mental illness including depression, anxiety, and addiction [49].

Interestingly, contrary to our results, LeMoult et al. carried out a meta-analysis that evaluated the degree of association between the so-called “early-life stress” (ELS) and the early onset of depression. Additionally, eight specific forms of ELS including poverty were analyzed to determine the risk of depression in children and adolescents. In this regard, the authors reported that, along with diseases or natural disasters, poverty did not demonstrate a significant association to the risk of depression in the early stages of life. They also indicated that although the effects of poverty on the health and intellectual development of children and adolescents have been documented, the emotional aspects have not been consistently evaluated. This is probably due to the disparity in the duration of the state of poverty in the samples studied or the lack of more specific details about the times of exposure, for example, during critical periods of childhood development [50].

Furthermore, researchers have shown the conditioning effects of low SES on development in different brain structures at early ages, particularly for DS [51]. Luby et al. demonstrated that children and adolescents who came from low SES had reduced hippocampal, amygdala and prefrontal cortex volume [51,52,53]. Interestingly, Barch et al. described changes in amygdala and hippocampal connectivity in preschoolers who experienced poverty and struggled with depression during school years [54]. These structures have been also associated to language acquirement and fullness, and even IQ has been reported to be significantly decreased in children from economically underprivileged families compared to those from privileged backgrounds [18]. Later, in adolescence, it has been reported that lower SES is associated with an increase in methylation of the proximal promoter of the serotonin transporter gene, which predicts greater increases in threatened amygdala reactivity. As is recognized, greater increases in amygdala reactivity moderate the association between a positive family history for depression and the later manifestation of DS [49].

Poverty affects the family in regard to the acquisition of nutritious foods, high-quality childcare, and safe/quality housing [21,22]. In terms of housing, housing complexes such as neighborhoods or condominiums are considered to provide shelter/security complexes such as neighborhoods or condominiums predestined to provide shelter/security against excessive climatic situations, pests, and environmental discomforts such as noise. However, housing is a primary factor influencing the development process, since it institutes, manages and maintains the environment for the physical, psychosocial, cultural and financial well-being, as well as, above all, the quality of life of people. Within the house, the physical structure, design and particularities such as floors, walls, and ceilings must be considered [55,56,57].

As previously mentioned, SES affects child poverty and, therefore, child development. In addition, community conditions and public services such as welfare and social services as well as healthcare are directly influenced by social and governmental policies [18,58].

In our research, no relationship was observed between the risk of children presenting DS associated with their families’ access to health services. It is important to mention that having the possibility of receiving mental health attention for children may be essential so that the symptoms of depression that were detected do not progress into chronic depression. Previously, Hynek et al. reported increasing odds (OR = 1.99, 95% CI 1.90–2.08) of using services of mental healthcare by young adults and adolescents with background of constant low parental income during preschool age [17].

Moreover, observations from last decades report that within each generation, the risk of depression occurrence at an earlier age increases. Onset at early ages also relates to worse clinical prognosis, functional affectation, and trend to chronicity [59]. Even worse is the fact that the Global Burden of Diseases, Injuries, and Risk Factors Study (GBD) 2019 reported that mental disorders persisted between the ten primary causes of global burden since 1990 [60]. Preschool depression has become a strong predictor of children continuing to have depression and meeting the criteria for full-blown depression at school age. When debuting during childhood and adolescence, depression adopts a chronic course with common recurrence, and a two to four times higher risk of having depression in adulthood [59]. For example, eating disorders and depression frequently co-occur during adolescence. Kenny et al. reported that the most important connecting symptoms of depression and eating disorders are the irritability sensation, the social eating, and the depression per se. It is noteworthy, as previously mentioned, according to the DSM-5, that irritability and anhedonia are clinical characteristics of early depression and are part of the ESDM 3-6 inventory [4,6,7]. Early detection of and attention to depressive symptoms during childhood can be crucial to prevent children from developing eating disorders. In addition, reports indicate that for suicidal ideations, the odds are five times greater in this type of patients with mental disorders, although, fortunately, the risk of attempting it is not increased. Suicidal ideations were seen in a higher proportion of adolescents with eating disorders [61].

Finally, it is important to approach the personality of individuals in order to understand the evolution of DS towards MDD. The Big Five Scale (McCrae and Costa, 2008) has helped describe types of personalities without statement of association or nature of behaviors and psychopathology [62]. Reports on adolescents and young adults point out that people with a high aperture (people who are motivated to seek new experiences constantly and engage in their own understanding of themselves), as well as with high neurotic scores (high anxiety, tension/constant worry, instability/emotional insecurity, tendency to feel guilty, etc.) have a higher risk of depression and increased severity of it at younger ages [63,64,65]. Less outgoing, less responsible and orderly and less independent personalities are also indicative of increased risk of early severe depression in children [63]. In fact, a personality profile has been proposed as a useful element to recognize students at risk of adverse emotional conditions to take preventive strategies; this might even help predict possible consequences for treatment planning [66,67].

As for the limitations of the study, the objectives were not aimed at analyzing the proportion of children who showed MDD determined by the ESDM 3-6 as a screening tool and defined psychiatric pathology (depression, anxiety, stress, etc.) by the specialist. It would be desirable to follow-up children diagnosed with DS and assess their biopsychosocial development. Finally, it was not possible to include all the children from the municipal schools because their parents did not agree to participate, or they could not attend the meetings where they were informed about the research. Likewise, it is important to consider additional social circumstances of the population in the future in order to have a more complete understanding of, for example, the phenomenon of migration from indigenous communities to this municipality.

Considering all the points addressed in the study, regarding perspectives, there are many other environmental and sociodemographic factors that have been associated with preschool depression. In regard to the environmental aspects, high levels of pollution, continuous contact with cigarette smoke, as well as social and maternal relationship with children, physical, mental and sexual violence, among others, stand out as particular triggering depression conditions [68,69,70,71]. As depression rates are rising among children, teens and adults of all ages, also exacerbated by the COVID-19 pandemic, these features would be very important to be acknowledged in future studies [72].

## 5. Conclusions

The great value of this research lies in the fact that screening tests were applied to a considerable semi-rural population in which DS could be identified in preschoolers. In this study, screening tests were adequate to identify DS in the children at risk and refer them to a health specialist. The specialist would then complete the diagnosis and determine the treatment of a possible incipient pathology, improving the quality of life of the population where this type of study was performed.

According to the results obtained, it was observed that malnutritional status, overcrowding and multidimensional poverty in the preschoolers living in a semirural community increase the risk of developing DS. The results showed that malnutrition is an independent risk factor for presenting DS, and it is of greater importance in the group of UW.

Other elements in the univariate analysis that contributed to the development of DS were the condition of poverty, quality of housing and overcrowding. Interestingly, when the multivariate study was performed, the quality of housing and access to healthcare ceased to be significant. This suggests that in the future, another type of analysis can be carried out to help evaluate this phenomenon. However, DS occurred predominantly in the group of children who had parents with less than 9 years of schooling.

From the above, we recommend ESDM 3-6 as a useful tool to detect DS, main characteristics of MDD (suicides, self-laceration, etc.), or important outcomes associated to DS in schoolers (poor academic performance and psychosocial problems). It is desirable that all preschoolers diagnosed with DS are followed up and their biopsychosocial development continuously evaluated.

## Figures and Tables

**Table 1 children-10-00835-t001:** Demographic, and nutrition status of the subjects (n = 695 participants).

Variables		Male				Female			
		Depressive Symptoms		Non-Depressive Symptoms		Depressive Symptoms		Non-Depressive Symptoms	
		(n = 68)	%	(n = 312)	%	(n = 41)	%	(n = 274)	%
Normal weight		(n = 39)	57.35%	(n = 249)	79.80%	(n = 25)	60.97%	(n = 216)	78.83%
	Age (years)	4.59 ± 0.88		4.71 ± 0.80		4.48 ± 0.87		4.63 ± 0.82	
UW		(n = 13)	19.12%	(n = 15)	4.80%	(n = 8)	19.51%	(n = 17)	6.20%
	Age (years)	4.15 ± 0.90		4.07 ± 0.70		4.00 ± 0.93		4.00 ± 0.50	
OW/OB		(n = 16)	23.53%	(n = 48)	15.38%	(n = 8)	19.51%	(n = 41)	14.96%
	Age (years)	4.63 ± 0.71		4.60 ± 0.82		4.75 ± 0.89		4.83 ± 0.83	

UW: Underweight; OW: Overweight/OB: Obesity.

**Table 2 children-10-00835-t002:** Relationship between sex and depression, malnutrition, and poverty condition.

Association	Chi-Square Test Value	df, N	*p*
Depression	3.10	1, 695	0.078
Malnutrition	0.05	1, 695	0.822
UW	0.05	1, 695	0.822
OW/OB	0.18	1, 695	0.670
Poverty condition	2.03	1, 695	0.154
Access healthcare	1.17	1, 695	0.2802

UW: Underweight; OW: Overweight/OB: Obesity.

**Table 3 children-10-00835-t003:** Univariate analysis of the associations between prevalence of depression identification, nutritional status, poverty condition, and availability of health services (OR and 95% confidence intervals).

Variables	Depressive Symptoms	Non-Depressive Symptoms	OR	IC 95%	*p*-Value
	N	N			
Malnutrition					
Absent	64	465			
Present	45	121	2.702 *	1.771–4.145	*p* < 0.001
UW	21	32	4.768 *	2.570–8.795	*p* < 0.001
OW/OB	24	89	1.959 *	1.175–3.324	*p* < 0.01
Poverty					
Present	31	107	1.779 *	0.9911–2.630	*p* < 0.05
Absent	78	479			
Housing quality					
Absent	9	25	2.020 *	0.9606–2.659	*p* < 0.05
Present	100	561			
Overcrowding					
Present	24	87	1.619 *	0.8989–4.433	*p* < 0.05
Absent	85	499			
Family type					
Monoparental	24	62			
Biparental	85	526	0.417	0.247–0.705	*p* < 0.19
Parental education					
Mother					
More than 9 years	28	344	0.245	0.154–0.389	*p* < 0.001
Less than 9 years	78	235			
Father					
More than 9 years	23	314	0.249	0.150–0.413	*p* < 0.001
Less than 9 years	65	221			
Health Services					
Present	101	523	0.660 *	0.3130 to 1.360	*p* < 0.05
Absent	8	63			

UW: Underweight; OW: Overweight/OB: Obesity. * Statistical significant difference between Depressive Symptoms and Non-Depressive Symptoms *p <* 0.05.

**Table 4 children-10-00835-t004:** Multivariate analysis of the associations between prevalence of depression identification, nutritional status, poverty condition, and availability of health services (OR and 95% confidence intervals).

Variables	Depressive Symptoms	Non-Depressive Symptoms	OR	IC 95%	*p*-Value
	N	N			
Sex					
Male	68	312	2.108	0.879–2.372	0.15
Female	41	274			
Malnutrition					
Absent	64	465			
Present	45	121	7.490	1.224–3.402	0.01
UW	21	32	9.480	1.527–6.719	0.01
OW/OB	24	89	2.131	0.856–2.897	0.144
Poverty					
Present	31	107	4.378	1.039–3.197	0.05
Absent	78	479			
Housing quality					
Present	9	25	0.434	0.401–3.399	0.77
Absent	100	561			
Overcrowding					
Present	24	87	4.869	1.080–3.660	0.05
Absent	85	499			
Family type					
Monoparental	24	62	8.826	0.224–0.736	0.01
Biparental	85	526			
Parental education					
Mother					
Less than 9 years	28	344	7.822	0.199–0.753	0.005
More than 9 years	78	235			
Father					
Less than 9 years	65	221	5.714	0.229–0.864	0.05
More than 9 years	23	314			
Health Services					
Present	101	523	3.005	0.892–6.449	0.08
Absent	8	63			

UW: Underweight; OW: Overweight/OB: Obesity.

## Data Availability

The data presented in this study are available on request from the corresponding author. The data are not publicly available due to the characteristics of the study.

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
