# Peer review of "Nutritional Status and Poverty Condition Are Associated with Depression in Preschoolers"

_children, 2023, doi:10.3390/children10050835_

Round 1

Reviewer 1 Report

Dear Authors

It is a writing that draws attention to little-known facts and relationships between mental health and the environment, it is also clear that it has been carried out by a work team with the necessary capacity to constantly improve, so despite not accepting its publication For all the reasons mentioned below, hopefully the research team will continue with the same heuristics, working on improving the aspects recommended below.

The writing is presented as an inductive investigation but it needs to be reaffirmed as such. At first it is presented as a case study, however as the text progresses it becomes a revisionist study. After the correlation between nutrition and depression, the inductive chain is lost and information is presented totally separate from the information obtained in the field.

The authors have taken the ESDM 3-6 test as the final criterion for establishing the diagnosis of depression, when its creators themselves establish that it is only an initial step, a sieve, from which progress must be made in the development of the diagnosis, taking advantage of different sources of information but above all the personal clinical relationship: Domènech-Llaberia, E. M. (1995). Depresión en la edad preescolar. En E. Domènech-Llaberia, Actualizaciones en psicologia infantil (0 a 5 años), 155-168. Barcelona: PPU.; Araneda, N., Moreno, C., Jané, M., & Domenèch-Llaberia, E. (1998). Características Psicométricas de la escala “The General Rating of Affective Symptoms for Preschoolers (GRASP)”: estudio preliminar en población preescolar española. Infanto – rev. Neuropsiq. da Inf. e Adol., 56-68.).

It is also very important that it is not mentioned that the ESDM 3-6 test has only recently been applied to Latin America and Mexico; but unfortunately they are not quoted, nor are references made to these investigations:

1. Reyna Y. Alejandra, (2005) Depresión infantil en los niños de 3-6años de edad en el jardín de niños dolores vega anza, tesis para obtener el grado de Maestra en Psicología Clínica, Universidad Autónoma de Durango.

2. Emmy Gálvez C.; Cinthia Medina G.; Manuel Fernández, (2011) Relación entre estado nutricional e indicadores de estabilidad psicológica. Un enfoque desde la salud con poblacion preescolar chilena. VII Congreso Nacional de Ciencias del deporte y educación Física. https://altorendimiento.com/relacion-entre-estado-nutricional-e-indicadores-de-estabilidad-psicologica-un-enfoque-desde-la-salud-con-poblacion-preescolar-chilena/

3. Pérez Álvarez Fidji Danaé (2019) Propuesta psicopedagógica para disminuir síntomas depresivos y favorecer el lenguaje- comunicación a nivel preescolar. Tesis Para obtener el Grado de Maestro en Psicología, Facultad de Psicología, maestría en psicología.   

 Another important issue is that the ESDM 3-6 instrument was created and applied in the city of Barcelona and one of the fundamental criteria for the validation of the tests is that it be established under similar development conditions or, where appropriate, the differences be analyzed, in light of the results obtained by closing the gaps or differences that may have arisen.

It is also necessary to say that in this Catalan proposal of the ESDM 3-6, the "General Rating of Affective Symptoms for Preschoolers" (Kashani, Holcomb & Orvaschel, 1986) was applied together with the ESDM 3-6, this last was not applied alone. So not only are the methods used as truth criteria not being met, but the scope of the results is being exceeded with respect to the original validation.

Even the fact that it is announced as something relevant that the same percentages have been obtained as in the study of the city of Barcelona, ​​even 20 years later in this rural community in central Mexico, far from being something positive is something negative. So much coincidence from such dissimilar times and places is not a good reference.

1. The sample of 695 preschoolers is not actually the source of the information but rather the teachers to whom the ESDM 3-6 instrument was applied. In this sense, it should be clarified how many teachers participated, how many students for each teacher was questioned.

2. It is not clear, where the information comes from to determine the housing conditions of each of the children.

3. It is not mentioned how old or what socioeconomic profile each of the participating boys and girls has.

In the text several statements are presented in relation to sexual differences without presenting sex as a variable in the research design or in the research approach. Furthermore, the results do not show any difference between both sexes, so the strangeness of the variable is very clear, both with respect to the approach and the results. In addition, references to sexual differences are indicated from the genetic level, an aspect not considered by the ESDM 3-6.

Likewise, the inclusion of neuro-scientific information is somewhat strange compared to the original approach, despite the fact that several lines are dedicated to it. It is completely decontextualized from the inductive research that was announced at the beginning. In other words, the neurological data is disconnected from the approach of an inductive investigation.

For all of the above, it is recommended to correct these errors before their possible publication.

Author Response

We are thankful with the referees for carefully reviewing the manuscript and the opinions regarding science and its presentation. In what follows, the referee’s comments are in italics, the author's responses are in blue, and the changes made are highlighted.

Reviewer 2 Report

In the manuscript, the authors conducted a study of Nutritional status and poverty condition are associated with depression in preschoolers. 

These comments are indicated in the text of the manuscript. For example, in the methodological part, one should indicate the qualitative characteristics of the objects of the sample (ratio male to female). From the tables with the results, the meaning of the control is not clear?

Author Response

(The authors gave the same response as above.)

Author Response

(The authors gave the same response as above.)

Reviewer 4 Report

Comments and suggestions

It was my pleasure to review this manuscript dealing with the nutritional status and poverty conditions associated with depression in preschoolers. This manuscript aims to examine the environmental (malnutrition) and social factors (poverty conditions, access to health systems, etc) for the development of depression in preschoolers. A cross-sectional study was conducted on Mexican children from municipal preschool institutes, and a questionnaire consisting of the Preschool Depression Scale for Teachers (ESDM 3-6), children’s medical history, and social demographic was given to their parents to be answered. In brief, I found the topic quite interesting. But with the sole objective of improving the quality of the manuscript, I will allow myself to make a few comments. In addition, I suggest this manuscript should undergo extensive English revisions. Because many minor mistakes exist and should be corrected.

Introduction part:               

1.      I suggest the introduction section needs to be revised precisely and structurally. The authors addressed a total of 12 paragraphs to describe the contents of the introduction; hence the contents look very uncentralized. Please restructure your contents and specifically addressed the prior findings or evidence of nutritional status, poverty conditions, and depression among preschoolers.

Materials and Methods

1.      This manuscript did not address how to design this study’s sampling and how to collect the data and studying process. The manuscript only mentioned that 981 children from municipal preschool institutes in Chiconcuac de Juarez, a semi-rural municipality of the State of Mexico were asked to take part in the study. This part should mention in the revised manuscript.

2.      It is not specified how the sample size was calculated to be representative. A margin of error of 5% and a confidence interval of 95% is normally accepted. The number of individuals that made up the study population was not expressed, nor was the minimum number of the necessary sample meeting the criteria I listed above.

3.      This study collected data through a questionnaire that consisted of an ESDM 3-6 scale and other important predictors such as poverty conditions and access to healthcare. What is the definition of poverty condition (no poverty vs. housing quality vs. overcrowding) and access to healthcare (With Health Services vs. without Health Services)? Did this study set up inclusion criteria or exclusion criteria? This part didn’t mention in the methods.

4.      Statistical analysis of this manuscript only adopts univariate analysis. Please add multivariate analysis that adjusts other confounding variables such as parents’ education and parents' marital status etc. in the revised manuscript in order to reveal real results.

Results

1.      This part only can be addressed and showed descriptive statistics. No future deep information can illustrate. Please add the results of the multivariate analysis.

2.      Page 4, lines 178-182:

In terms of poverty condition per se (OR = 1.779, 95% CI [0.9911 - 2.630]; n=31); as well as the components analyzed in the present study such as housing quality (OR = 2.020, 95% CI [0.9606 - 2.659] ; n=9) and overcrowding = 1.619, 95% CI [0.8989 - 4.433]; n=24) may rise depressive symptoms. Also, access to healthcare was negatively associated with the risk of presenting DS (OR = 0.660, 95% CI [0.3130 to 1.360] ; n=8). In fact, these results did not achieve statistically significant differences because all results of 95% confidence interval contain 1. Moreover, these results only adopted univariate analysis. Please revise it.

3.      Page 5, table 3:

Please confirm and check the results in table 3, especially for the percentage of malnutrition (78.6% vs. 21.24%). Please explain how to calculate it.   

Discussion

1.      Page 5, lines 198-207:

You mentioned that “Mexican government has defined poverty in a multidimensional analysis considering aspects of the living environments of the population from three areas: economic well-being (needs associated with the goods and services that can be acquired through income), social rights (deficiencies in the use of their rights for community development) and the territorial context (degree of social organization, as well as others reflected for social development)”. Does your study adopt the same concept to collect your data on poverty conditions?

2.      page 6, lines 236:

Please correct the word “fullfill” into “fulfill”.

3.      Page 6, lines 247:

You addressed that “Accordingly, Lewis et al. found that females between 4 and 14 years were predisposed to an increasing path of DS than males at the same age.[14]”. Please check the punctuation marks of the sentence.

4.      You mentioned that “the design of strategies to promote mental healthcare, good nutritional habits, and generation of social programs to prevent/reduce poverty is urgent since these developmental stages are vital to have healthy adolescents and adults in an integral way”. This manuscript examined the nutritional status and poverty conditions associated with depression in preschoolers. Does your study have any practical and specific policy implications in this area? Please explain your opinion and add this important information to the discussion part.

Author Response

(The authors gave the same response as above.)

Round 2

Reviewer 3 Report

The recommendations have been strictly adhered to, and an excellent job has been performed.

Reviewer 4 Report

The authors have carefully responded to the comments and added valuable points to the revised manuscript.  The reviewer accepts the present manuscript. Thank you.